# WHAT A DIFFERENCE A PIXEL MAKES: AN EMPIRICAL EXAMINATION OF FEATURES USED BY CNNS FOR CATEGORISATION

## ABSTRACT

Convolutional neural networks (CNNs) were inspired by human vision and, in some settings, achieve a performance comparable to human object recognition. This has lead to the speculation that both systems use similar mechanisms to perform recognition. In this study, we conducted a series of simulations that indicate that there is a fundamental difference between human vision and CNNs: while object recognition in humans relies on analysing shape, CNNs do not have such a *shape-bias*. We teased apart the type of features selected by the model by modifying the CIFAR-10 dataset so that, in addition to containing objects with shape, the images concurrently contained non-shape features, such as a noise-like mask. When trained on these modified set of images, the model did not show any bias towards selecting shapes as features. Instead it relied on whichever feature allowed it to perform the best prediction – even when this feature was a noise-like mask or a single predictive pixel amongst 50176 pixels. We also found that regularisation methods, such as batch normalisation or Dropout, did not change this behaviour and neither did past or concurrent experience with images from other datasets.

## 1 INTRODUCTION

Object recognition in humans is largely a function of analyzing shape (Biederman, 1987; Hummel, 2013). A wealth of data from psychological experiments show that shape plays a privileged role in object recognition compared to other diagnostic features such as size, colour, luminance or texture. For example, Biederman & Ju (1988) showed that error rates and reaction times are virtually identical in a recognition task when full coloured photographs of objects are replaced by their line drawings even when colour was a diagnostic feature. This indicates that shape-based representations mediate recognition. Similarly, Mapelli & Behrmann (1997) found that, for patients with an object recognition deficit (visual agnosia), surface colour played minimal role in aiding object recognition unless the shape of the object was ambiguous, indicating that shape is instrumental to recognition, whereas surface characteristics such as colour and texture play only a secondary role. More recently, Baker & Kellman (2018) have shown that participants extract shape information automatically from arrays of dot patterns within the first 100ms of stimulus onset, even for tasks where extracting this information may be detrimental to performance on a task. Experiments from developmental psychology show that this privileged status of shape starts early in life and becomes stronger with age. For example, Landau et al. (1988) found that 2-3-year-old children as well as adults weight shape more heavily than size or texture when generalising the name of a learnt object to novel instances. They also found that the weight placed on shape increases in strength and generality from early childhood to adulthood. Following Landau et al. (1988), we will call this privileged status of shape in performing recognition a "shape bias".

By contrast, it is unclear whether shape plays a privileged role in how convolutional neural networks (CNNs) categorise objects. It is often claimed that CNNs learn representations of objects that are similar to the representations that monkeys and humans use when identifying objects (Rajalingham et al., 2018), and that CNNs largely rely on learning shape representations in order to categorise objects (Kubilius et al., 2016; Ritter et al., 2017; Jozwik et al., 2017). On the other hand, there are a growing number of studies that show that CNNs often categorise images on the basis on non-shape attributes of images. This is demonstrated by the existence of adversarial images that are

confidently classified as a familiar category despite the lack of any shape information in the input (Nguyen et al., 2015), adversarial images that contain the correct shape but altered colours that are confidently misclassified (e.g., categorizing an image of an airplane as a dog when only the colour of the plane has been manipulated), and large reductions in performance when trained coloured images are converted to greyscale (Geirhos et al., 2017) or the colours are inverted (Hosseini et al., 2017). In addition, there are demonstrations that CNNs can easily learn to categorise random patterns of pixels that have no shape (Zhang et al., 2016). All of these findings suggest that shape may not play a privileged role in CNN's object categorisation, or that the relevant role of shape and non-shape features depends on the specific model or training conditions.

Here we systematically explore the impact of non-shape features in the categorisation performance of convolutional neural networks on CIFAR-10 images. We introduced non-shape features to images by adding informative noise-like masks to the training set. We tried several types of masks and an extreme version where the non-shape feature consisted of just a single pixel with a location correlated to the image category (see Figure 1 and Appendix B). We show that CNNs often learn and depend on non-shape features that are highly diagnostic of object categories and often fails to learn anything about shape under these conditions. This highlights that CNNs simply picks up whatever statistical structure is most relevant to learning the training set, with shape playing no special role. Note that this does *not* imply that CNNs do not encode shape information under any circumstance, but that shape does not seem to be weighted more than other diagnostic features, even when these features are noise-like masks or the luminance of a single pixel. Importantly, this behaviour contrasts with humans, for whom shape plays a privileged role in performing recognition, even in the presence of much more salient diagnostic features such as size, colour or texture. If the models are to more closely capture human performance, these results suggest that additional machinery needs to be added to networks in order to prioritize the role of learning shape-based representations while performing the object categorisations. This might also reduce the model's susceptibility to being fooled by non-shape features of images, and being more robust to various forms of non-shape noise that currently reduce performance.

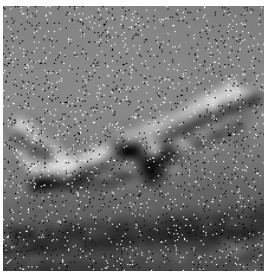 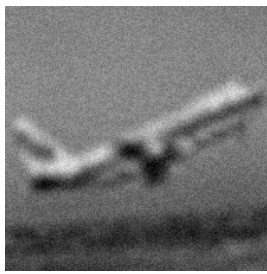 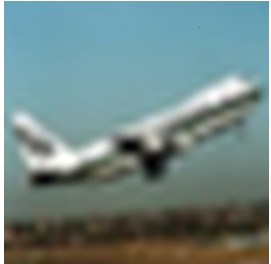

(a) Salt-and-pepper noise     (b) Additive noise     (c) Single diagnostic pixel

Figure 1: Hidden in plane sight. Images taken from CIFAR-10 dataset and scaled up to 224x224 pixels. (a) Image is converted to greyscale and we add a salt-and-pepper noise-like mask to each training image; (b) Image is converted to greyscale and we add uniform additive noise mask to each training image; (c) A single diagnostic pixel is inserted in the image (top-right, though nearly invisible to the naked eye).

## 2 EXPERIMENT PREMISE & DESIGN

We modified the CIFAR-10 dataset (see Appendix A) so that each image contained not only features that pertain to the shape (e.g. object outlines) but also features without any shape information. As non-shape features we used noise-like masks that were combined with the original image. Two different types of masks were used: the *salt-and-pepper noise mask* turned a certain proportion of image pixels to either black or white, while a *additive uniform noise mask* added a value sampled from a uniform distribution to each pixel of an image. We also tested an extreme form of the salt-and-pepper noise mask where only one pixel was turned to a particular colour. In this case the location and colour of the pixel were different for different categories but correlated for images within a category. Masks were independently sampled for each category but were either fixed for all images

in a category (in which case the mask predicted the category) or sampled from a distribution with category-dependent parameters (in which case these parameters predicted the category). So these modified images concurrently contained features that were related to shape and features without shape information. Both types of features were predictive of the category of an image. Examples of modified images are given in Appendix B.

We trained the model on these modified sets of images and tested it under three conditions. During the 'Same' condition, the test set was modified in exactly the same manner – i.e., either images in each category were generated by using the same mask as that for the training images of that category (when the mask was fixed) or they were generated by using the same parameters as the parameters used to generate noise masks for training images of that category (when the mask was variable). In contrast, during the 'Diff' condition, the noise masks (or their parameters) for each category were swapped with another category. The premise here was that if the model based it's decisions on shape-related features, then it would ignore the noise mask and the performance during 'Same' and 'Diff' condition should be similar. On the other hand, if the model relied on properties of the (non-shape) mask, then it's performance would be worse in the 'Diff' condition compared to the 'Same' condition. Finally, we used a third, 'NoPix', condition to estimate the extent to which the network relied on features of the noise mask. In this condition, we presented the network with a version of the image without any mask, with the premise that the difference between the performance in 'Same' and 'NoPix' condition should quantify the relative extent to which the network relied on shape-based and non-shape features. We ran all of the simulations using the well-known VGG-16 network (Simonyan & Zisserman, 2014) and checked that our main results replicate for a deeper network, ResNet-101 (He et al., 2016). To give the model the best chance to recognise shape-based features, all simulations were carried out on CNNs that had previously been trained on ImageNet categories and replaced only the fully-connected layers to perform the new classification task. We then turned the learning rate to a small value and trained these networks on the new classification task (see Appendix C for simulations were learning was completely frozen in convolutional layers).

## 3 RESULTS

In the first set of experiments, all images in a category had the same noise mask. For salt-and-pepper mask, this meant that noise masks were sampled independently for each category, but the same set of pixels in each image were modified for all images in a category. Similarly, for the additive uniform noise mask, the same mask was added to each image in a category. For the single pixel noise, the location and colour of the added pixel were independently sampled for each category, but kept constant for all images in a category.

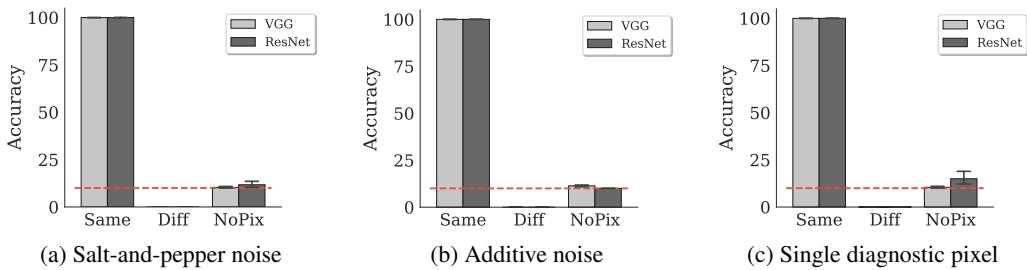

(a) Salt-and-pepper noise        (b) Additive noise        (c) Single diagnostic pixel

Figure 2: Accuracy on test images under the three types of noise-like masks shown in Figure 1. 'Same': the noise-like mask has same properties for test and training images of each category; 'Diff': the properties of the mask during test are swapped with another category from training; 'NoPix': No mask is inserted. The dashed (red) line indicates chance performance and error bars show 95% confidence interval. Light and dark gray bars show accuracies on the `VGG-16` and `ResNet-101` convolutional neural networks, respectively.

The results of these experiments are shown in Figure 2. We obtain the same pattern of results for all three cases: when noise mask in the test images matches the noise mask in training images, the model classifies images nearly perfectly; when noise masks are swapped, the accuracy drops to zero;

when the mask is completely removed, the categorisation accuracy is at chance. Furthermore, we get the same pattern of results on both VGG and ResNet networks. These results clearly indicate that the model learns to completely rely on features of the noise-like mask, rather than any shape-related information present in the images. Even in the extreme case, where only one pixel amongst 50176 was diagnostic of the category, the model prefers to classify based on this feature over other shape-related features present in each image.

One response to this preference for these non-shape, but statistically relevant, features is that it can be addressed by regularisation methods that prevent overfitting to idiosyncratic features in the training dataset. The network is simply failing to select the more general feature – shape – choosing instead to memorise the noise-like mask. If this was the case, then using some of the well-known regularisation methods, such as *Batch Normalization*, *Weight Decay* or *Dropout*, may diminish this preference for the non-shape feature. However, we found that regularisation did not make any significant difference to the results above (Figure 3). Even when a dropout of 50% was added after every convolution layer, the pattern of results remained the same.

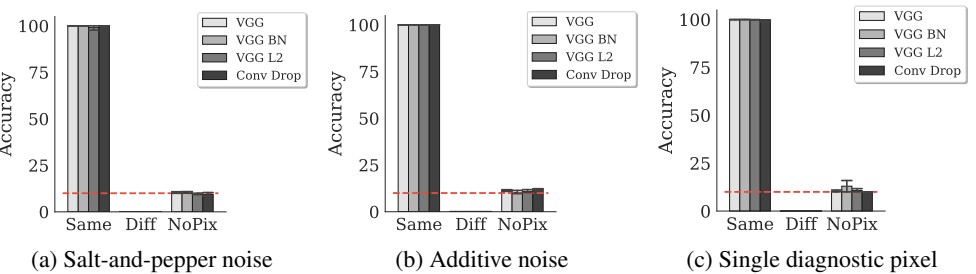

Figure 3: Effect of regularisation. Graphs show accuracy for Same, Diff and NoPix conditions using models with four types of regularisations. The base model, VGG-16, uses dropout in fully-connected layers but not after convolutional layers; VGG_BN: VGG-16 with Batch Normalisation; VGG_L2: The 16-layer VGG CNN trained with weight-decay of $1e-3$; Conv Drop: A seven-layer CNN with dropout after every convolution and fully connected layer.

Another possible reason why humans prefer to rely on shape-related features to categorise objects while CNNs do not is that humans are guided by past experience and bring this past knowledge to new categorisation tasks. So when a human sees an object with superimposed noise, they generalise from past experience and look for shape-based information, paying less attention to non-shape related features such as the noise-like mask in above images. In order to test whether the network will similarly generalise from experience on other tasks, we conducted two further experiments.

In the first experiment, we divided the training set into two subsets. The first subset ('with pix') contained three randomly chosen categories from CIFAR-10 and, like above, contained a category-correlated pixel in all images of these categories. The second subset ('unaltered') contained the remaining seven categories from CIFAR-10 and was left unaltered – i.e. we did not add the category-correlated pixel to images of this subset. The network was then trained on all ten categories at the same time. We were interested in finding out whether the network generalised from one subset to another and started using the features used to categorise images in the 'unaltered' subset to images of the 'with pix' subset. The results from this experiment are shown in Figure 4a. The model learnt to predict the images in the 'unaltered' subset with nearly 90% accuracy. However the performance on the 'with pix' subset still completely depended on the location and colour of the added pixel: accuracy was nearly 100% when test images contained the pixel in the same location, but dropped below chance when this pixel was removed. Thus, the network did not seem to generalise the features learnt in the 'unaltered' categories to the categories containing the diagnostic pixel.

The above experiment required the model to *simultaneously* learn on categories with and without a diagnostic pixel. In a second experiment we tested what happens when the network is first trained on images that did not contain such a pixel (a 'before' phase) followed by a second ('after') phase in which such a pixel was inserted in the training set. This setup more closely resembles the situation faced by humans, where they bring their knowledge from previous categorisation tasks and use this knowledge to select the shape-based features to a new task.

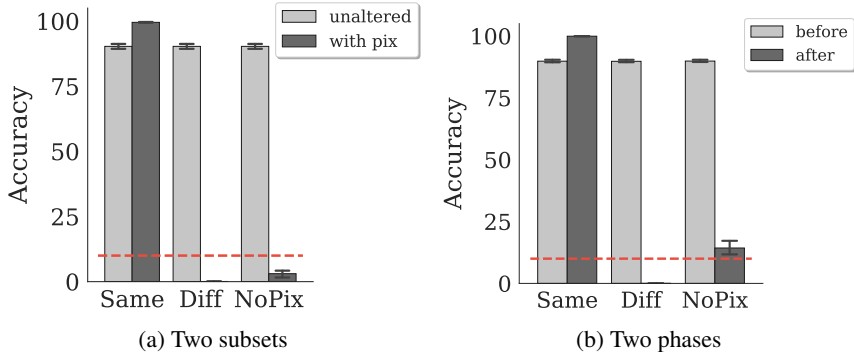

Figure 4: Lack of generalisation. Accuracy under Same, Diff and NoPix conditions for (a) two subsets: an 'unaltered' subset where no noise-like mask was inserted in training images and a 'with pix' subset where a single diagnostic pixel was inserted, and (b) for two phases: a 'before' phase, where a pre-trained VGG network was trained on images without any noise masks and tested on the three conditions, and an 'after' phase, where the model from before phase was then trained on images with a single diagnostic pixel.

In the first phase, a pre-trained VGG network was trained on an unaltered CIFAR-10 training set. Once the network had learnt this task, we trained it on the modified set of images in a second phase, introducing a predictive pixel in each category. So all that changes between the 'before' and 'after' phases is the insertion of a single category-correlated pixel to each image. We observed that (Figure 4b), instead of relying on past experience with these images, the model learnt to completely rely on the predictive pixel to perform categorisation – accuracy dropped from nearly 100% to 0% between 'Same' and 'Diff' conditions. Crucially, the model completely forgot about how to perform categorisation when the predictive pixel was removed – accuracy was close to chance in the 'NoPix' condition during the 'after' phase. Thus learning about the diagnostic feature seemed to be accompanied by unlearning previously learnt representations. This, catastrophic forgetting, is a well-known problem in neural networks (McCloskey & Cohen, 1989) and contrasts with how humans transfer their knowledge from one task to another.

## 3.1 Introducing variability

The non-shape features used in the experiments above have all been completely invariant from one image to another within a category. It can be argued that these features are selected by the model over other shape-based features because they provide a very strong predictive signal and consequently suppress the selection of any shape-based features. It is possible that if these features contained larger variance, the model would be more likely to rely on shape-based features while performing categorisation. In a series of experiments where we increased the variability of the non-shape (noise-like) features, we noticed that this was generally not the case – the model still relied a lot more on these features than on any shape-based features to perform categorisation.

The first type of variability we introduced was to sample the noise mask independently from a distribution for each training and test image within a category. In order to make these noise masks diagnostic of an image's category, a parameter of this distribution correlated with an image's category. For the salt-and-pepper noise, this meant that the probability, $p$, of changing a pixel to black or white was different for each category. Thus, the parameter, $p$, became diagnostic of the category. However, the masks now varied from image to image and were independently sampled with the (category-dependent) probability, $p$. Similarly, for the additive uniform noise, masks could vary from one image to other within a category but the mean of the distribution depended on each category (see Appendix A for details). For the single diagnostic pixel, the inserted pixel could vary in location from one image to the other, but was generated from a Gaussian distribution with a mean determined by the category of the image and a fixed standard deviation. Similarly the colour of the pixel was sampled from a Gaussian distribution with a mean determined by the category of the image and a fixed standard deviation.

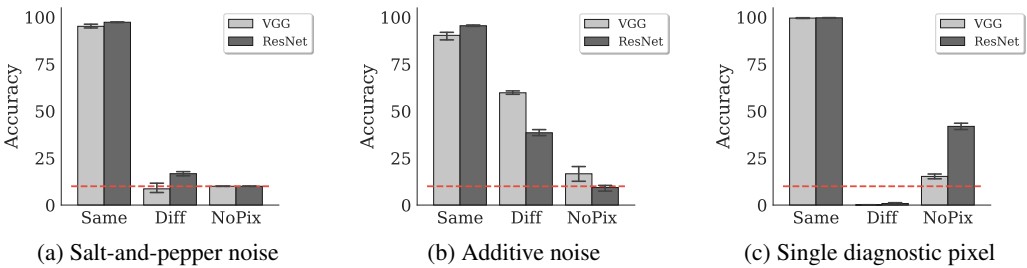

(a) Salt-and-pepper noise    (b) Additive noise    (c) Single diagnostic pixel

Figure 5: Accuracy on test images when the noise mask varies between images of a category. Training images contain (a) salt-and-pepper noise, or (b) additive uniform noise, or (c) just one diagnostic pixel. The dashed (red) line indicates chance performance. See Figure 2 for a description of the 'Same', 'Diff' and 'NoPix' conditions.

The results of introducing a variable noise mask are shown in Figure 5. Introducing variability in the location and colour of the single diagnostic pixel brought very little change to the VGG model's behaviour (compare Figure 5c with Figure 2c). Performance in the NoPix condition was somewhat better for ResNet, however the pattern of result remained the same – performance dropped substantially from the Same to NoPix condition. Similarly, introducing variability in the salt-and-pepper masks lead to only a minor change in behaviour of the model, with accuracy in 'Diff' condition dropping to chance, rather than $0\%$. The most intriguing change in behaviour occurred when variability was introduced to the additive uniform noise mask (Figure 5b). While the VGG and ResNet networks differed quantitatively in these results, the pattern of results remained the same: when the noise mask was completely removed (NoPix condition) the model performed *worse* than when the images contained a noise mask from a different category (Diff condition). In other words, removing the mask makes the image less informative for the model, not only compared to images with the correct category-correlated (Same) mask, but also compared to images with the incorrect (Diff) mask – the model seems to rely on the presence of noise to make an inference.

Next, we examined how the model changes it's behaviour when only a subset of images contain a diagnostic non-shape feature. We restricted this experiment to the case of a single diagnostic pixel. The location and colour of this pixel were fixed across all images of a category, but we introduced stochasticity in the presence of this pixel within a training image. Figure 6 shows the change in accuracy for the 'NoPix' condition with an decrease in the probability with which a pixel is present in a training image. We specifically focus on the 'NoPix' condition as the accuracy on this condition is inversely correlated with how much the network relies on this pixel to predict the output category.

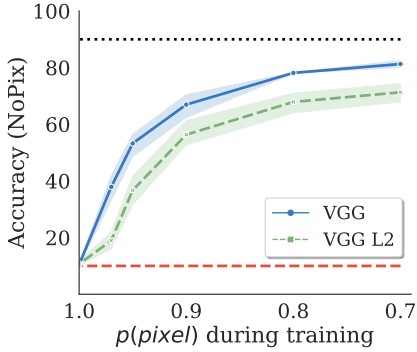

Figure 6: Accuracy of the model on images containing no mask, as a function of the fraction of training images containing a diagnostic pixel. The solid (blue) and dashed (green) lines plot this relation for a network trained without and with weight-decay, respectively. Dashed (red) line at the bottom shows chance performance. Dotted (black )line at the top shows performance of a network trained on images without any noise mask.

It can be seen from this figure that accuracy increases smoothly, rising sharply at first and then slowing down as the probability of the pixel being present in a training image decreases. This smooth increase is consistent with the hypothesis that the learning algorithm selects the feature based on the predictive power of the feature; as the single pixel becomes less predictive, the network starts relying on other features to choose the output category. This smooth increase also indicates that the model is able to combine information from this diagnostic pixel with other features that it uses to predict the output category. For example, when the pixel is present in only 90% of the images, the model is able to correctly categorise an image containing no diagnostic pixel 70% of the time, indicating that it simultaneously represents both the diagnostic pixel as well as other features that it uses to perform categorisation.

The figure also shows that a single pixel present in training images adversely affects the performance of the network even when it is present on only a fraction of the images. When the network is trained on images from the original CIFAR-10 dataset, it's accuracy is close to 90% (dotted black line); inserting a single pixel on 70% of the images meant that the performance decreased by more than 10%. This reduction in performance could be due to one of two reasons: (a) the network mistakes a pixel value in one of the original images as a predictive pixel and performs an incorrect classification based on this pixel, or (b) adding a mask to a subset of training images means that the network has to learn the correct classification function on a fraction of the original dataset. This decrease in the size of the dataset may be affecting its performance. Further testing will be needed to correctly establish which of these reasons is responsible for a reduction in performance. Lastly, we also observed that L2 regularisation made the performance of the network worse on the original images when a diagnostic pixel was inserted on a fraction of the images. While L2 regularisation should help the network learn a more general solution, in this case it lead to the opposite effect.

## 3.2 TRANSLATION INVARIANCE VERSUS PIXEL LOCATION

While conducting the above experiments, we noticed that the model managed to discriminate between categories based simply on the location of a single pixel – i.e. even when the colour of the inserted pixel was same for all categories, the network was able to perfectly classify images containing this diagnostic pixel. This implies that the network was able to represent the location of the inserted pixel, which is surprising considering CNNs are designed to achieve translation invariance through a combination of convolution and pooling operations. In our final experiment, we wanted to understand how the network represents the location of the inserted pixel and uses it to perform categorisation.

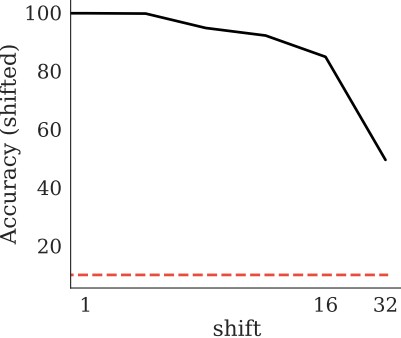

Figure 7: Accuracy during the 'Same' condition, when location of the pixel in the test set is *shifted* compared to the training set. The amount of shift was varied between 1pixel and 32pixels (shown on $log_2$ scale). Dashed red line shows chance performance and the shaded region shows 95% confidence intervals.

In order to do this, we modified the training dataset so that the colour of the inserted pixel was the same (black) across all categories, but it's location depended on the category. Moreover, for the test images, we inserted the pixel at a location which was *shifted* between 1 and 32 pixels (along the x-axis) with respect to the training images. We varied the degree of this shift and observed how the accuracy of classification changes as a function of this shift. If the network memorises the

exact location of each pixel, the slightest shift should lead to a catastrophic decay in categorisation accuracy. In contrast, we noticed that the network was reasonably robust to a shift in the location of the test pixel and performance only degraded significantly when the location of the test pixel was shifted by sixteen pixels or more (Figure 7). Across simulations we noticed that this inflection point depended on the relative location of the diagnostic pixels for different categories; the network seemed to be dividing the image into different decision regions based on the relative location of the diagnostic pixel. When the diagnostic pixel was shifted by more than a certain amount, the image was classified as a different category. These boundaries seemed to be discrete – i.e. the classification did not gradually change from one category to another as the pixel was varied. Rather, all the images were classified as one category or the other based on the location of the pixel.

## 4 RELATED WORK

Su et al. (2017) demonstrated that CNNs trained on CIFAR-10 and ImageNet can be fooled by introducing a single adversarial pixel, with error rates of $68\%$ and $41\%$, respectively. Unlike our approach the model was trained with uncorrupted images and the authors systematically searched for an adversarial pixel that lead to any sort of error (so-called non-targeted attack). So, in contrast to our goal, the goal of their study was not to explore whether CNNs systematically learn non-spatial information. However, the findings are in line with ours – the CNNs trained by them do not seem to be categorising based on shape. Rather, it must be that there was, by chance, some pixel value that was highly correlated with a given output category and the model picked up on this idiosyncratic correspondence. As a consequence, when this pixel was added to another category the model was fooled.

Geirhos et al. (2017) reported that CNNs are not as robust as humans to different types of noise. Note that they use the term "noise" differently to us. Our use of the term "noise-like mask" refers to non-shape features that are diagnostic of an image's category, while Geirhos et al. (2017) insert non-diagnostic noise in their stimuli. They found that training CNNs to become robust to one distribution of noise did not make the CNNs to be robust to other forms of noise. That is, the models found it easy to learn about specific forms of noise, just as we observed above. As the authors note, this is problematic given that the space of possible image distortions is vast and it is not possible to train CNNs to be robust to all possible forms of noise. We would suggest a key solution to this observation is to introduce some sort of shape-bias to CNNs that might make networks more robust to a wide range of noise rather than training on the noise per se.

## 5 CONCLUSIONS

In a series of simulations we found that the VGG network trained to categorise CIFAR-10 images that included noise-like masks diagnostic of the output categories often learned to categorise on the basis of these masks rather than the CIFAR images themselves. Indeed, the models often entirely relied on the masks, and performed at floor when the noise was removed from the images. Even though we specifically engineered our dataset to contain non-shape features, it is well-known that popular datasets such as CIFAR and ImageNet contain various biases due to conditions under which the images were captured as well as the different motivations for construction of the datasets (Torralba & Efros, 2011). Our results suggest that CNNs may be relying too heavily on non-shape features when categorising images and therefore may be extremely susceptible to non-shape biases present within datasets. This, in turn, could be the source of various idiosyncratic behaviours such as being confounded by fooling images (Nguyen et al., 2015) or being overly sensitive to colour (Hosseini et al., 2017), noise (Geirhos et al., 2017) or even single pixels in images (Su et al., 2017). This contrasts with human visual object recognition that is largely based on shape (Biederman, 1987). It will be important to introduce shape biases to CNNs if they are to mirror human object recognition performance more closely. The introduction of shape biases may also prove useful in making CNNs more robust to various non-shape manipulations of images (e.g., changes in colour or the introduction of noise) that often impair performance.

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

## A   IMAGE PREPROCESSING & MODEL DESCRIPTION

We used a method similar to Geirhos et al. (2017) to transform images from the CIFAR-10 dataset (https://www.cs.toronto.edu/~kriz/cifar.html). All transformations were performed using the Pillow fork of the Python Imaging Library (https://pillow.readthedocs.io). Each 32x32 pixel image was rescaled to 224x224 pixels using the `PIL.Image.LANCZOS` method. For the salt-and-pepper and additive noise masks, each image was transformed from RGB to greyscale using `PIL.Image.convert()` method. For the extreme case of single pixel, the images were not colour transformed (we obtained qualitatively similar results if images were transformed to greyscale). When images were transformed to greyscale, their contrast was adjusted to $80\%$ by scaling the value of each pixel using the formula: $0.8 \times v + \frac{1-0.8}{2} \times 128$, where $v$ was the original value of the pixel in the range $[0, 255]$.

The salt-and-pepper mask was created by taking the transformed greyscale image and setting each pixel to either black or white with a probability $p$. When the mask was fixed for a category, all images had the exact same set of pixels that were turned either black or white and the $p$ was set to 0.05. When the mask varied from image to image within a category, the pixels were sampled independently for each image and the probability $p$ was fixed for each category but varied between categories in the range $[0.03, 0.06]$.

The additive uniform noise mask was created by taking the transformed greyscale image and adding a value sampled from the uniform distribution $[-w, w]$ to this image, where $2w$ was the width of the uniform distribution and was set to 8. When the noise mask was fixed, this sampling was done only once per category and the same mask was added to each image. When the mask was variable, it was sampled independently for each image from a distribution $[\mu - w, \mu + w]$, where $\mu$ was the mean that depended on the category and varied in the range $[-50, 50]$.

The single pixel mask was created by choosing a random location, $(x, y)$, (sampled from a uniform distribution on the interval $[0, 224]$) on the image and changing the colour of the pixel to a value $c$ (sampled from a uniform distribution on the interval $[0, 255]$). When the mask was fixed for each category, $(x, y, c)$ remained constant for all images in a category, but varied between categories. In other words, the pixel was inserted at different locations and was of different colours for different categories, but all images within a category had the pixel at the same location and of same colour. When the mask was variable, each of $x, y$ and $c$ were sampled independently for each image from a Gaussian distribution with a constant variance and a mean that depended on the category of the image. If any value in a sampled set of $(x, y, c)$ values fell out of their respective range, that value was re-sampled.

All simulations reported in this study (except for the Conv Drop simulation in Figure 3) were carried out using a pre-trained 16-layer VGG network (Simonyan & Zisserman, 2014) provided by the `torchvision` package of `Pytorch`. This network had been pre-trained on the ImageNet dataset. We replaced the fully-connected layers of this pre-trained model with three fully-connected layers with Dropout after the first two layers. This model was then trained on the modified training set using the RMSProp gradient descent optimization algorithm (see Ruder, 2016) with learning rate of $1e-5$, a momentum of 0.9 and a cross-entropy loss function. We also experimented with Adam (Kingma & Ba, 2014) and results remained qualitatively same. For testing the effect of Dropout in the early layers, we constructed a six-layer convolutional neural network with three convolutional layers and three fully connected layers and dropout after every convolutional layer. The same learning rule and parameters were used as for the VGG network and we experimented with several model architectures with most architectures giving similar results. This network was able to achieve an accuracy of $70\%$ on categorising the CIFAR-10 dataset. The input to both types of networks was a 3-channel RGB image. For greyscale images, all three channels were set to the same value.

## B    EXAMPLE IMAGES

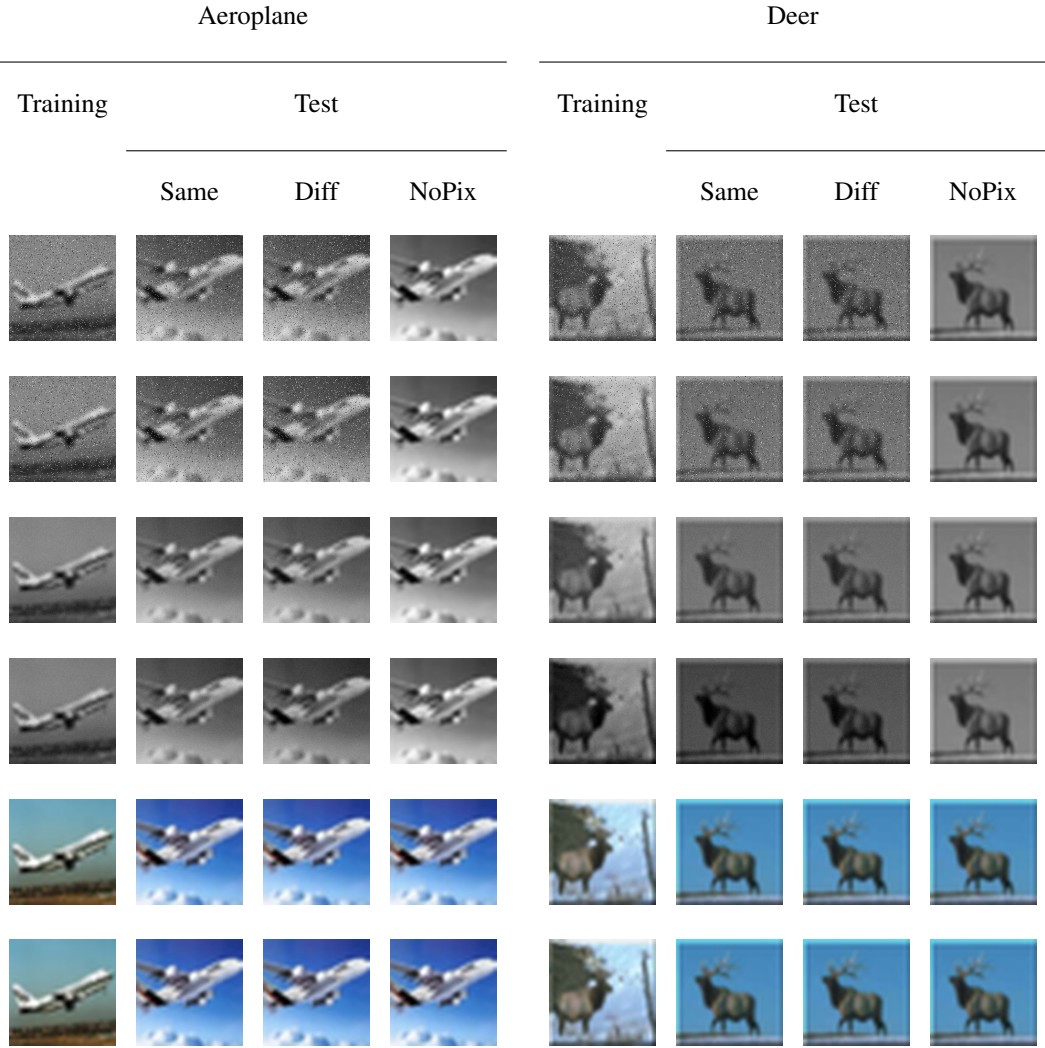

Figure 8: Examples of images used for training and testing. The columns show the condition under which the image was used and the rows show the type of noise-like mask inserted. These noise masks are, respectively, (row 1) salt-and-pepper noise with a fixed mask, (row 2) salt-and-pepper noise with a variable mask, (row 3) additive uniform noise with fixed mask, (row 4) additive uniform noise with a variable mask, (row 5) single diagnostic pixel, fixed location and colour and (row 6) single diagnostic pixel with variable location and colour.

## C    EFFECT OF FREEZING LEARNING

A trick that is frequently used in deep learning networks to enable them to generalise between tasks is to switch off learning completely in the convolution layers after pre-training on one dataset. The assumption here is that different categorisation tasks share low level features used for classification. Switching off learning means that these features are not unlearned as a result of training on a new dataset. We noticed that when this is done, the network was less sensitive to the introduction of a category-correlated pixel (Figure 9, light gray bars). This is not surprising – freezing the weights in convolution layers prevents the network from learning any new features such as the single predictive pixel. In fact, even under this extreme constraint, the performance of the network is significantly affected by the presence of a single diagnostic pixel. The effect of introducing additive uniform noise are even more acute (dark gray bars in Figure 9) – accuracy drops to nearly 40% in the 'Diff' and 'NoPix' conditions, showing that the model learns to rely on the noise-like mask even when learning in convolution layers is switched off.

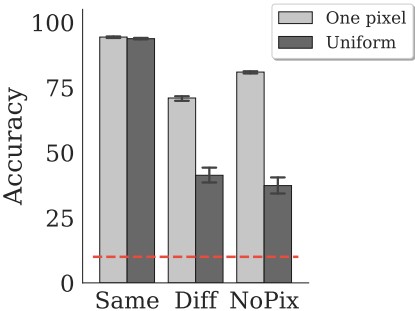

Figure 9: Effect of completely freezing convolution layers. Accuracy on images containing either Additive (Uniform) noise-like mask or single diagnostic pixel (One pixel) when the learning rate for convolution layers is set to zero after pre-training on ImageNet.

