# OpenReview forum: "What a difference a pixel makes: An empirical examination of features used by CNNs for categorisation"
_ICLR.cc/2019/Conference_

### Official Review · AnonReviewer3 · 2018-11-02
**clever experiments with interesting implications**

**Rating:** 7
**Confidence:** 5

**Review:**

This paper adds to a growing body of literature which suggests that modern CNNs use qualitatively different visual strategies for object recognition compared to human observers. More specifically, the authors create shapeless object features (by adding noise masks in various forms or single pixels that are predictive of categorization to object images) to study how much CNNs rely on shape information (as humans would) as opposed to shapeless arbitrary statistical dependencies between pixels.

The hypotheses tested are straightforward and the experiments cleverly answer these questions. On the negative side, there is nothing groundbreaking in this study. As acknowledged by the authors, the results are not all that novel in light of recent work that has already shown that one could conduct adversarial attacks by corrupting a single pixel as well as work that has shown that CNNs do not generalize to noise degradations they have not seen. Still, there is value in the work presented as the empirical tests described address the role of shape in object recognition with CNNs.

In a sense, the present study offers a null result and obviously, the work would have been much more significant had the authors offered a mechanism to get CNNs to learn to prioritize "shape" features (then verifying that such network would work on CIFAR, but performed poorly on the shapeless images).

Additional analysis involving visualization methods to further explain why shape features were ignored would have been a plus– with bonus points for providing a heuristic to determine the "shapelessness" of a convolution kernel.

---

> ### Author Response · Authors · 2018-11-19
> **Response to review: novelty**
>
> We thank the reviewer for his/her positive comments.  We agree with the reviewer that the results may not be ground-breaking as a demonstration of the limitations of CNNs, but as the reviewer correctly identifies, the key novelty of our work lies in contrasting how shape is treated very differently by humans and CNNs. In this sense, our study is more constructive than a typical adversarial attack, as it points in a direction where deep learning research could benefit from understanding the representations and processes underlying human vision. (Please see the revised Introduction, which highlights this contrast).
>
> Another contrast with adversarial studies (such as single pixel attacks) is that we manually insert non-shape features in the training set, rather than exploring susceptibilities of CNNs trained on well-known datasets. By creating these biases in the dataset, we were able to understand how such biases affect performance and the surprising extent to which non-shape features can drive performance (even in cases when such non-shape features would be nearly imperceptible to a human being). Also, our study is the first to systematically manipulate the conditions in which non-shape information impacted performance: We varied the type of non-shape noise, we varied the timing at which the noise was introduced (and found that the non-shape information overwrote shape information through catastrophic interference), manipulated the percentage of images in which the noise was embedded, and varied the form of regularization in order to see whether the effects are related to overfitting (we found no effect).  This is a very different approach than past papers that search for adversarial images, but our findings may help explain why some adversarial images are effective.
>
> We completely agree that it is an important question about how to make CNNs focus more on shape (that is, how to induce a shape bias in CNNs).  This might help make these models less susceptible to adversarial images, and, make the models more informative about human vision.

---

### Official Review · AnonReviewer1 · 2018-11-06
**Shape-bias or shortcut-bias and catastrophic forgetting?**

**Rating:** 4
**Confidence:** 4

**Review:**

The paper seeks to establish via a series of well-designed experiments that CNNs trained for image classification differ in a fundamental way from human vision – they don’t encode shape-bias like human vision. Towards this goal, the authors modified the training data with ‘shortcut’ features to be functions of the category label using single diagnostic pixels and their placements, noise masks (salt and pepper, additive) and their parameters and demonstrate that image categorization CNNs learn whatever statistical features are there in the data most relevant to the learning task.

Investigation of the properties of neural architectures like CNNs and using the understanding thus developed to create better neural architectures, learning algorithms and training paradigms are good directions for the community and from that perspective, the direction explored in the paper is of great relevance and interest to the community.

The paper presents careful experimentation to establish that image categorization CNNs learn the statistical features most relevant to the learning task. And, it seems to satisfactorily demonstrate this. It shows that such features could be single pixels, noise masks and even parameters of stochastic distributions which randomly produce these features, as long as the parameters are predictive of the image category. The experiments are well designed and they demonstrate this point quite well. They also demonstrate the well-known problem of catastrophic forgetting.

Nonetheless, there are significant drawbacks in the presented work:

1.	The experiments don't seem to effectively demonstrate the main claim of the paper that categorization CNNs do not have inductive shape bias (encode shape information). (Let’s make this claim more concrete: categorization CNNs when trained via supervised learning with paired training data of {(image, category_label)} do not have inductive shape bias.)

The best way to demonstrate this would have been to subject a trained image-categorization CNN to test data with object shapes in a way that the appearance information couldn’t be used to predict the object label. The paper doesn’t do this. None of the experiments logically imply that with an unaltered training regime, a trained network would not be predictive of the category label if shapes corresponding to that category are presented.

2.	Due to the surprising results (especially the intensity of observed effects), we tried to reproduce some results from the paper in our lab and faced difficulties in doing so:

a.	We tried to replicate Figure 4(a) 'nopix' and 'same' cases on a standard setting (VGG-12-BN on CIFAR-10). The results deviated significantly (33%-72% margin) on ‘nopix’ case from the results reported in the paper on a much stronger setting (1/3072 pixels vs 1/50176 as in the paper). Please let me know any crucial settings (see below) that we might have missed.

Details: We used the vgg-cifar10 repository by chengyangfu. The only additions was fixing the pixel values while sending in the data. The code is anonymized and hosted here: https://file.io/qiziAK. The pixel values in CIFAR-10 using the pytorch dataloader are between [-0.45, 0.45] theoretically, typically much smaller. We set the (0,0) RGB pixels categorically spacing it uniformly from [-0.25, 0.25), [-0.025, 0.025), [-0.0025, 0.0025) as a simple experiment. The third case did not suffer any decrease in the nopix case or any increase in the pix at all. The first case showed significant deviations from the claimed results with the no-pix resulting in ~43% accuracy which is 33% off vis-à-vis the results in the paper. The ‘same’ setting didn’t achieve 100% either though it got close - achieving 98.4%.

Summary: The paper presents an important line of investigation to understand the properties of CNNs. However, it fails to effectively demonstrate its main claim. Further, we had difficulties in reproducing the results. As it stands, the submission is not of publishable quality.

I encourage the authors to do more careful experimentation to demonstrate their main claim and perhaps work on strategies to encourage CNNs to learn more meaningful features, including ‘shape’-features and submit to a future conference.

Revision: Updated my rating to acknowledge that the reproducibility issue is addressed.

---

> ### Author Response · Authors · 2018-11-19
> **Response to review: Replication**
>
> We thank the reviewer for some really useful feedback and especially the effort put in to replicating our study. We are pleased that this reviewer found our findings sufficiently surprising that he/she ran a replication study based on Figure 4(a) and provided us a link to determine exactly what was done.  We were of course concerned to hear that our findings did not replicate.  However, it turns out that this is due to the way in which the dataset has been generated. When generated in the correct manner, our results indeed replicate using the code provided by the reviewer. The other major concern the reviewer had was whether our experiments correctly examine our main claim. We address both these concerns in detail below.
>
> [Replication] There are two crucial differences between our code and the reviewer’s in the way the pixel was inserted into any image. Firstly, we inserted the pixel at a different (x,y) location for each category. This made the pixel location diagnostic of the category of each image. In contrast, the reviewer inserted the pixel at the same location (0,0) for all categories, making the location of the pixel non-diagnostic. Secondly, the reviewer assumes that the pixel values in CIFAR-10 using the pytorch dataloader are between [-0.45, 0.45]. This is not the case in the code provided by the reviewer. Due to normalisation, these values are, in fact, approximately between [-2.5, 2.5]. Therefore, pixel values used by the reviewer to test the results ([-0.25, 0.25] / [-0.025, 0.025] / [-0.0025, 0.0025]) provide a very weak diagnostic signal to the network.
>
> To check whether these settings make a difference, we modified the code provided by the reviewer in two ways: (i) we picked the (x,y) location of the pixel for each category randomly from a uniform distribution [0, 32) (but kept it constant for all images within a category), and (ii) we picked each of the RGB values for the inserted pixel from a uniform distribution in the range [-2.0, 2.0), [-1.0, 1.0) or [-0.01, 0.01). Other than these changes the code provided by the reviewer remains the same (only changes are around lines 60-70 and then 175-180). When the pixel values were in the interval [-2.0, 2.0), the accuracy is ~100% in the Same condition and between 10-20% in the NoPix condition, which is very close to what we find. The small remaining difference could be due to a difference in learning algorithm used (we used RMSProp while the reviewer used SGD) or due to pretraining (we used a VGG-16 network pretrained on ImageNet while the reviewer used a VGG-11 network that was trained from scratch on the modified dataset). When pixel values are in the range [-1.0, 1.0) we again get an accuracy of ~100% in the Same condition, which drops to ~20% in the NoPix condition. Even when the inserted pixel provides a very weak diagnostic signal, with pixel values nearly at the mean [-0.01, 0.01), the network shows a large drop in performance from ~100% in the Same condition to ~50% in the NoPix condition, clearly demonstrating the reliance on this diagnostic pixel. Furthermore, it should also be noted that even under the conditions used by the reviewer, where all the categories had the pixel inserted at the same location and pixel values were grayscale and in the small range [-0.25, 0.25), performance dropped from 98% to 42% when the pixel was removed. It is inconceivable that this will happen for human participants and provides additional support for our observation that the model simply picks up whatever statistical structure is most relevant to learning the training set, with shape playing no special role.
>
> We have uploaded the modified code and log files here: http://s000.tinyupload.com/?file_id=24861367338244091333. However, we do understand that some of these settings may not have been completely obvious in the previous version of the manuscript. To facilitate future replication, we have moved the description of some of these settings from the Appendix to the main text. Furthermore, in order to check if the results replicated on other networks, we have now run the key simulation using ResNet-101 and found that we get a similar pattern of results. We have updated the key figures (Figure 2 & Figure 6) to show these results.

---

> > ### Author Response · Authors · 2018-11-19
> > **Response to review: Claim**
> >
> > [Claim] The reviewer was also concerned that we do not effectively demonstrate the main claim of the paper “that categorization CNNs do not have an inductive shape bias (encode shape information)”. In fact, this is not our claim. We agree that, given the right dataset, CNNs may indeed encode shape information. The reviewer is right in saying that our study does not demonstrate that CNNs do not encode shape information when trained on well-known datasets such as CIFAR-10 and ImageNet. If we wanted to test this, the experiment suggested by the reviewer would indeed make sense. However, our intention was to test a different hypothesis: that shape has a privileged status amongst features used to perform recognition. It is this privileged status of shape that is frequently referred to as "shape-bias" in psychological literature. There is a lot of evidence from psychological experiments showing that humans show a preference for using shape over other features such as size, colour or texture when any of these features are diagnostic of an object’s category. (We have added examples of these studies to the revised manuscript). We suspected that CNNs do not have such a shape bias, and in order to test how far this lack of shape bias could be pushed, we created a dataset where, in addition to shape, an almost imperceptible feature (a single pixel / an additive noise mask) was also diagnostic of an image’s category. Our experiments show that even for these nearly imperceptible features, the model did not show any preference for using shape. While this does not mean that CNNs do not use shape when trained on any given dataset, they do show that they tend to pick on whatever feature is more predictive, a behaviour that contrasts with human cognition.
> >
> > Given the reviewer’s comment, we can see that this aspect of the paper was not clearly laid out. Therefore, we have revised the manuscript, clarifying our claim and laying out the psychological literature that motivated our study at the outset.

---

> > > ### Comment · AnonReviewer1 · 2018-11-26
> > > **Claims and revised claims**
> > >
> > > Thanks for the reply. I actually do like the direction of this work. However, in it's current form, it needs a major revision.
> > >
> > > 1. CNNs and human cognition: In my mind, no one in the ML or the CNS community seriously speculates that CNNs and human cognition (way more complex than CNNs -- the current CNNs are most likely not even bio-plausible) use similar mechanisms and should show the same behavior. This speculation might (perhaps) exist in the psychology community which may treat both as 'black' boxes to be probed with well thought-out experiments to understand their behavior. So, to me, the main driving motivation is not nuanced.
> > >
> > > 2. CNNs and inductive bias: I also feel that the way the term CNNs is used lacks nuance. ('CNNs' are not all the same architecture and even if some categorization CNNs are shown to be deficient, so what?) By 'CNNs', the authors likely mean the architecture, the training task including the dataset, the loss encoding the task, the optimization algorithm and it's details, put together. The inductive shape bias is likely to be a function of all the above.
> > >
> > > A reasonable expectation is that a large capacity network with a large diverse training data trained on a large set of diverse and representative training tasks should induce an inductive shape bias.
> > >
> > > In any case --
> > >
> > > (a) The authors agree that the experiments don't show whether CNNs encode shape bias or not.
> > >
> > > (b) I agree that the privileged nature of the shape feature in human cognition is indeed something that CNN models should emulate and aspire for. There is no expectation that they current do -- a milder statement that categorization CNNs do not learn to represent shapes well is hardly a surprise. Further, vis-a-vis the experiments, please note that shape is a high level abstracted feature while pixel intensities are not and the fact that CNNs do not privilege shape (if they don't need to) is hardly surprising .
> > >
> > > However, even this claim is weakly demonstrated by the paper.
> > >
> > > (b-1) As shown by Figure 9, revised manuscript, pretrained (frozen) convolution layers are more robust to appearance perturbations introduced in the data. However, even here, the task-dependent layers are trained. The fact that the convolution layers 'leak' the perturbation signal through is surprising -- though this will happen if all information is encoded in one way or the other in the features at the output of the CNN layers. If this happens, the shortcut-bias will be learnt - this is not surprising.
> > >
> > > (b-2) Comparing with shape bias in a human cognition and saying something about 'CNNs' seems a stretch. I'm not well versed in the psychology literature so I'm not sure of the following. A psychological study would likely test a very well trained human cognition system which already has shape bias learnt on a large amount of diverse data over a long period of time with hybrid supervision on multiple hierarchical tasks (I'm being very hand-wavy here!). How much of this is retrained by psychology experiments is not pointed out in the paper. Comparing them to experiments on CNNs where there is a shortcut-bias in training (even if training only the final layers of the CNNs) is not particularly revealing and I'm not sure what meaningful statements can be made from such a comparison.
> > >
> > > So, to summarize, I'm fine with an empirical examination of CNN features, an enunciation of the takeaways and the future directions for learning representations from data using the currently used architectures and learning paradigms, etc.. However, I think the whole comparison with human cognition is a distraction. The current manuscript needs a major revision and can't be accepted in the current form.
> > >
> > > I'll revise my rating to reflect that the reproducibility issue is addressed.

---

> > > > ### Author Response · Authors · 2018-11-28
> > > > **Comparisons between CNNs and biological vision**
> > > >
> > > > In the first round of reviews, the reviewer raised two main criticisms that were both incorrect. Now the reviewer has raised a new main criticism (“that no one in ML or the CNS community seriously speculates that CNNs and human cognition… use similar mechanisms”) that is again mistaken.  The reviewer seems to be rejecting our paper on the basis that it is not interesting to compare CNNs with human vision, in spite of the fact that ICLR explicitly invites research on deep learning methods applied to neuroscience and computational biology, and in spite of the fact that vision is the domain of cognition that is most frequently modelled using neural networks.
> > > >
> > > > In the first round, the reviewer was sufficiently surprised by our results that he/she attempted to replicate our findings and reported that they did not replicate.  In response, we not only showed that our results replicate, but also showed that the same findings are obtained using a different architecture, ResNet-101.  In the second round, the reviewer finds our demonstration of the lack of shape bias unsurprising, claiming there is no reason to expect CNN to relate to human vision. We admit to finding this new major criticism to be frustrating because it is factually incorrect.  We point to many references below to research in neuroscience, psychology, and machine learning making comparisons between CNNs and biological vision at the level of both behaviour and mechanism.  In a separate comment, we respond in detail to all the points in the second review.  The numbering of the responses corresponds to the numbering used by the reviewer (so R1 is the response for reviewer comment 1, etc.).

---

> > > > > ### Author Response · Authors · 2018-11-28
> > > > > **Studies comparing CNNs and human cognition**
> > > > >
> > > > > Cadieu, C. F., Hong, H., Yamins, D. L., Pinto, N., Ardila, D., Solomon, E. A., ... & DiCarlo, J. J. (2014). Deep neural networks rival the representation of primate IT cortex for core visual object recognition. PLoS computational biology, 10(12), e1003963.
> > > > >
> > > > > Cichy, R. M., Khosla, A., Pantazis, D., Torralba, A., & Oliva, A. (2016). Comparison of deep neural networks to spatio-temporal cortical dynamics of human visual object recognition reveals hierarchical correspondence. Scientific reports, 6, 27755.
> > > > >
> > > > > Dodge, S., & Karam, L. (2017, July). A study and comparison of human and deep learning recognition performance under visual distortions. In Computer Communication and Networks (ICCCN), 2017 26th International Conference on (pp. 1-7). IEEE.
> > > > >
> > > > > Eickenberg, M., Gramfort, A., Varoquaux, G., & Thirion, B. (2017). Seeing it all: Convolutional network layers map the function of the human visual system. NeuroImage, 152, 184-194.
> > > > >
> > > > > Elsayed, G. F., Shankar, S., Cheung, B., Papernot, N., Kurakin, A., Goodfellow, I., & Sohl-Dickstein, J. (2018). Adversarial examples that fool both human and computer vision. arXiv preprint arXiv:1802.08195. (Now published in NIPS 2018)
> > > > >
> > > > > Geirhos, R., Janssen, D. H., Schütt, H. H., Rauber, J., Bethge, M., & Wichmann, F. A. (2017). Generalisation in humans and deep neural networks. Advances in Neural Information Processing Systems 31.
> > > > >
> > > > > Güçlü, U., & van Gerven, M. A. (2015). Deep neural networks reveal a gradient in the complexity of neural representations across the ventral stream. Journal of Neuroscience, 35(27), 10005-10014.
> > > > >
> > > > > Khaligh-Razavi, S. M., & Kriegeskorte, N. (2014). Deep supervised, but not unsupervised, models may explain IT cortical representation. PLoS computational biology, 10(11), e1003915.
> > > > >
> > > > > Kim, J., Ricci, M., & Serre, T. (2018). Not-So-CLEVR: learning same–different relations strains feedforward neural networks. Interface Focus, 8(4), 20180011.
> > > > >
> > > > > Kriegeskorte, N., & Douglas, P. K. (2018). Cognitive computational neuroscience. Nature neuroscience, 1.
> > > > >
> > > > > Kubilius, J., Bracci, S., & de Beeck, H. P. O. (2016). Deep neural networks as a computational model for human shape sensitivity. PLoS computational biology, 12(4), e1004896.
> > > > >
> > > > > Rajalingham, R., Schmidt, K., & DiCarlo, J. J. (2015). Comparison of object recognition behavior in human and monkey. Journal of Neuroscience, 35(35), 12127-12136.
> > > > >
> > > > > Rajalingham, R., Issa, E. B., Bashivan, P., Kar, K., Schmidt, K., & DiCarlo, J. J. (2018). Large-scale, high-resolution comparison of the core visual object recognition behavior of humans, monkeys, and state-of-the-art deep artificial neural networks. Journal of Neuroscience, 38(33), 7255-7269.
> > > > >
> > > > > Yamins, D. L., Hong, H., Cadieu, C. F., Solomon, E. A., Seibert, D., & DiCarlo, J. J. (2014). Performance-optimized hierarchical models predict neural responses in higher visual cortex. Proceedings of the National Academy of Sciences, 111(23), 8619-8624.
> > > > >
> > > > > Wallis, T. S., Funke, C. M., Ecker, A. S., Gatys, L. A., Wichmann, F. A., & Bethge, M. (2017). A parametric texture model based on deep convolutional features closely matches texture appearance for humans. Journal of vision, 17(12), 5-5.

---

> > > > > ### Author Response · Authors · 2018-11-28
> > > > > **Detailed responses**
> > > > >
> > > > > R1. Contrary to the reviewer's belief that "no one in the ML or CNS community speculates that CNNs and human cognition use similar mechanisms", this is an active area of research with studies finding evidence both of similarities and systematic differences between the two. Please see below for a (non-exhaustive) list of studies published, in recent years, in high-impact journals, in computational neuroscience, psychology and machine learning that compare human cognition/vision/neuroscience and CNNs.
> > > > >
> > > > > R2.  The reviewer claims that our use of the term CNN lacks nuance.  It is true that we haven't tested all possible CNN architectures, training sets, loss functions and optimization algorithms. It would be unfeasible to do this. However, we have selected some representative architectures and training sets carefully. We chose to test VGG-16 (and ResNet-101) because these architectures are known to perform at a comparable level to human object categorisation and have been frequently used for making comparison to biological vision (see references below). We chose ImageNet (for pre-training) and CIFAR-10 (for the pixel task) for similar reasons. Moreover, as far as we can test, our results generalise to a host of different architectures (we tested a four-layered CNN, VGG-11, VGG-16, Resnet-101), a variety of different optimisation algorithms (SGD, RMSProp, Adam) and a variety of different training regimes (ImageNet, CIFAR-10 or trained from scratch on the modified dataset). It is, of course, possible that we are incorrect in assuming that our results generalise. That is why we have made it clear in the manuscript exactly how we trained and tested the models and invite future research showing if there are conditions under which CNNs can show shape bias.
> > > > >
> > > > > Related to this, the reviewer also writes: “A reasonable expectation is that a large capacity network with a large diverse training data trained on a large set of diverse and representative training tasks should induce an inductive shape bias.” Can the reviewer point us to any data underlying his belief that a large enough network with a large training set induce shape bias?
> > > > >
> > > > > R2(a). The reviewer writes: “The authors agree that the experiments don't show whether CNNs encode shape bias or not.”
> > > > >
> > > > > No - we don't. We agreed that CNNs may be capable of encoding shape. This is not the same as having a shape *bias*. As far as we can see, there is no preference for shape compared to other features, hence no shape bias.
> > > > >
> > > > > R2(b) The reviewer's statement that "categorization CNNs do not learn to represent shapes well is hardly a surprise" goes against previously published work that suggests that CNNs perform categorization based on shape (e.g., Kubilius et al, 2016). This has led to the expectation that CNNs capture essential mechanisms underlying human vision. For example, Kriegeskorte & Douglas (2018) argue that "internal representations of deep convolutional neural networks provide the best current models of representations of visual images in inferior temporal cortex in humans and monkeys". Given these comparisons being made between CNNs and biological vision makes our results both interesting and important.
> > > > >
> > > > > R2(b-1).  The reviewer writes that the shape bias claim “… is weakly demonstrated by the paper” because “pretrained (frozen) convolution layers are more robust to appearance perturbations introduced in the data” as shown in Figure 9.
> > > > >
> > > > > It is not surprising size of the effect reduces (though it's still significant) when there is no learning in convolution layers.  In this case they retain their previous representations and don't rely entirely on the single pixel or noise mask. The reviewer complains that we still allowed learning in the fully connected layers. If we froze the learning in fully connected layers, then learning will be frozen in the *entire* network. How could we then train it on a fresh set of images?
> > > > >
> > > > > R2(b-2).  The reviewer points that humans and CNNs learn under very different conditions.  It is true that humans and CNNs undergo very different training conditions. Nevertheless, there is evidence (and we have cited it in the manuscript) that humans show such a shape-bias from a very early age, suggesting that it is not an artefact of the training environment. Furthermore, in order to see whether training makes a difference, we have designed a set of different studies to gauge whether pre-training on a different dataset makes a difference (it does not), learning simultaneously on a set of categories without a diagnostic pixel makes a difference (it does not), learning beforehand on CIFAR-10 makes a difference (it does not), whether a variation in the diagnostic noise makes a difference (it makes very limited difference). Still there may be training conditions that could be responsible for shape-bias and we invite future research to point this out.

---

> > ### Comment · AnonReviewer1 · 2018-11-19
> > **Replication verification**
> >
> > Unfortunately, we are not able to download the file -- we just get a download.php with 0 bytes. However, we took the values provided in the response above and are indeed able to verify that we are now able to reproduce the results in the paper (we did not test all experiments).
> >
> > Based on the above, our concerns regarding the replicability are met.  We thank the authors for the clarifications, and are satisfied with the corresponding changes made in the manuscript.

---

> > > ### Author Response · Authors · 2018-11-20
> > > **New link**
> > >
> > > Thank you for the quick response. We are pleased to hear that you have been able to replicate the findings. Apologies about the problem with the link. In case you're curious, here's a new link that should work: https://ufile.io/9iw95

---

### Official Review · AnonReviewer2 · 2018-11-10
**This paper addresses an important issue but fails to propose a solution**

**Rating:** 4
**Confidence:** 4

**Review:**

Humans leverage shape information to recognize objects. Shape prior information helps human object recognition ability to generalize well to different scenarios. This paper aims to highlight the fact that CNNs will not necessarily learn to recognize objects based on their shape. Authors modified training images by changing a value of a pixel where its location is correlated with object category or by adding noise-like (additive or Salt-and-pepper) masks to training images. Parameters of such noise-like masks are correlated with object category. In other words if one learns noise parameters or location of altered pixel for each object category, they can categorize all images in the training set. This paper shows that CNNs will overfeat to these noise based features and fail to correctly classify images at test time when these noise based features are changed or not added to the test images.

Dataset bias is a very important factor in designing a dataset (Torralba et al,. 2011). Consider the case where we have a dataset of birds and cats. The task is image classification. All birds' images have the same background which is different than cats' background. As a result the network that is trained on these images will learn to categorize training images based on their background. Because extracting object based features such as shape of a bird and bird's texture is more difficult than extracting background features which is the same for all training images.

Authors have carefully designed a set of experiments which shows CNNs will overfeat to non-shape features that they added to training images. However, this outcome is not surprising. Similar to dataset design example, if you add a noise pattern correlated with object categories to training images, you are adding a significant bias to your dataset. As a result networks that are trained on this dataset will overfeat to these noise patterns. Because it is easier to extract these noise parameters  than to extract object based features which are different for each image due to different viewpoints or illumination and so on.

This paper would have been a stronger paper if authors had suggested mechanisms or solutions which could have reduced dataset bias or geared CNNs towards extracting shape like features.

Antonio Torralba and Alexei A. Efros. Unbiased look at dataset bias. In Proceedings of the 2011 IEEE Conference on Computer Vision and Pattern Recognition (CVPR '11).

---

> ### Author Response · Authors · 2018-11-19
> **Response to review: clarification of contribution**
>
> We thank the reviewer for taking time to read the paper and for their feedback. The reviewer’s key concern was that he/she did not find our results sufficiently novel or surprising as it is already well established that bias in datasets can lead to incorrect generalisation, as shown by Torralba & Efros (2011).
>
> We agree that we are not the first to make this point.  Our contribution is to highlight the surprising extent to which non-shape features can drive performance, and indeed, even the slightest bias (even a single pixel) is enough for CNNs to ignore shape and rely on the diagnostic signal.  Also, unlike previous studies, we systematically manipulated the conditions in which non-shape information impacted performance:  We varied the type of non-shape noise, we varied the timing at which the noise was introduced (and found that the non-shape information overwrote shape information through catastrophic interference), manipulated the percentage of images in which the noise was embedded, and varied the form of regularization in order to see whether the effects are related to overfitting (we found no effect).   Most importantly, we manipulated the degree to which the noise biased the training set (e.g., we manipulate the fraction of images containing the noise mask and the variability in the mask from one image to another), and showed that CNNs are strongly impacted across all levels of noise (and types of noise).  This is important given that all image datasets undoubtedly included uncontrolled noise that is correlated with the output categories (as pointed by Torralba & Efros).  Our findings highlight that this may have a larger impact on performance than previously assumed (and indeed, may help explain how single-pixel attacks can be successful).
>
> Furthermore, our findings with CNNs contrast with human visual perception where the extraction of shape occurs quickly and automatically and shape holds a privileged status compared to other diagnostic features, such as size, colour or texture. These features may allow humans to overcome biases present in their own environments. As Torralba & Efros point out: “a human learns about vision by living in a reduced environment with many potential local biases and yet the visual system is robust enough to overcome this.” Being biased towards finding shape may be a way in which the visual system overcomes one type of dataset biases (ones due to non-shape features present within the environment), and our results show that this shape-bias is missing from CNNs.
>
> We disagree with Reviewer 1 that our findings are due to overfitting.  Rather than overfitting, we have shown that CNNs are happy to fit to non-shape data. This is why regularisation methods such as weight-decay, batch normalisation and dropout have no impact on the results (Figure 3).
>
> In light of the reviewer’s comments, we have revised the Introduction and Discussion to make these issues clear.

---

### Public Comment · (anonymous) · 2018-11-02
**Question regarding interpretation**

Thank you for your work, I have read it with great interest!

I have a question concerning the interpretation of your results. When CNNs are given the choice to learn shape features or a category-specific noise mask / pixel, you show that CNNs take the easy route and rely on the predictive noise pattern. You interpret this as a "fundamental difference" to human vision (which relies on shape); and contrast it to claims about human-like CNNs in the literature. These claims are usually being made about CNNs trained on normal, uncorrupted images (i.e., images without predictive noise patterns).

If you say that "CNNs do not have a shape bias", do you refer to CNNs in general (including CNNs trained under normal circumstances, i.e. on noise-free images) or only to CNNs trained with category-specific noise (as investigated in your work)?

---

> ### Author Response · Authors · 2018-11-19
> **Thank you for the important question**
>
> Thank you for reading the manuscript and asking the question. This is indeed an important point and related to a common misunderstanding about our work. We do not make the claim that CNNs do not learn about shape under any circumstance. Rather, we claim that CNNs do not show a preference towards learning shape, given other diagnostic features, even when these features are noise-like masks or even single diagnostic features. This behaviour contrasts with human vision. We have clarified this point in a revision of the manuscript, where we have discussed the psychological literature related to this shape bias. Please see our common response to above for further details.

---

### Author Response · Authors · 2018-11-19
**Response to All Reviewers (and the anonymous commenter)**

We would like to thank the reviewers for their positive comments and constructive feedback.  We have uploaded a revision in response to these points.  Here we respond to the most important concern shared by the reviewers, and then respond to the more specific comments under each review.

The main concern is that we have altered the CIFAR images by introducing noise strongly correlated with the output categories. Accordingly, we have only shown that CNNs rely on non-shape cues under (unusual) conditions in which non-shape information is highly diagnostic of the categories. On this view, there is no reason to assume that CNNs trained on unperturbed images will rely on non-shape information, and thus no reason to conclude that CNNs ignore shape, nor claim that CNNs fail to show a shape bias.

This concern reflects a misunderstanding.  When we conclude that CNNs do not have a shape bias we do not mean to imply that CNNs ignore shape.  Rather, we are claiming that CNNs do not have an inbuilt inductive bias to rely on shape and that they will use any sort of statistical regularities that can be exploited (shape or non-shape).  In our simulations we made non-shape features more diagnostic than shape, and accordingly, the CNNs used non-shape.   By contrast, there is a large literature in psychology that shows that humans do have an inductive bias to rely on shape when identifying objects, and accordingly, psychological theory predicts that humans would continue to rely heavily on shape even when non-shape features are more diagnostic of object category.  We are making a psychological claim that CNNs are importantly different than humans rather than claiming that CNNs ignore shape in general.

Looking over the paper again we see that we were not very clear on this point.  Our revision now includes a modified introduction with a more detailed discussion of the shape bias in human object recognition.  We now emphasize that our main question is whether CNNs have a preference to categorize images according to shape (a shape bias, consistent with psychological theory), or whether CNNs are equally happy to rely on non-shape information for the sake of object identification (and use whatever is more diagnostic in a given dataset).  We think this is an interesting question given that CNNs were inspired by the visual cortex and given that some researchers claim that CNNs capture important properties of human vision (e.g., Rajalingham et al., 2018). In order to accommodate all of these changes and still stay within the page-limit, we have moved one of the experiments that was not central to the main story to Appendix C.

Rajalingham, R., Issa, E. B., Bashivan, P., Kar, K., Schmidt, K., & DiCarlo, J. J. (2018). Large-scale, high-resolution comparison of the core visual object recognition behavior of humans, monkeys, and state-of-the-art deep artificial neural networks. Journal of Neuroscience, 38(33), 7255-7269.

---

### Meta-Review · Area_Chair1 · 2018-12-10
**metareview: fundamentally flawed approach**

**Confidence:** 5
**Recommendation:** Reject

**Metareview:**

This paper claims to demonstrate that CNNs, unlike human vision, do not have a bias towards reliance on shape for object recognition. Both AnonReviewer1 and AnonReviewer2 point to fundamental flaws in the paper's argument, which the rebuttal fails to resolve. (AnonReviewer1's criticisms are unfortunately conflated with AnonReviewer1's reluctance to view neuroscience or biological vision as an appropriate topic for ICLR; nonetheless AnonReviewer1's technical criticism stands).

These observations are:

AnonReviewer2:

"Authors have carefully designed a set of experiments which shows CNNs will [overfit] to non-shape features that they added to training images. However, this outcome is not surprising."

AnonReviewer1:

"The experiments don't seem to effectively demonstrate the main claim of the paper that categorization CNNs do not have inductive shape bias"

"The best way to demonstrate this would have been to subject a trained image-categorization CNN to test data with object shapes in a way that the appearance information couldn’t be used to predict the object label. The paper doesn’t do this. None of the experiments logically imply that with an unaltered training regime, a trained network would not be predictive of the category label if shapes corresponding to that category are presented."

The AC agrees with both of these observations. CNN behavior is partially a product of the training regime. To examine the scientific question of whether CNNs have similar biases as human vision, the training regimes should be similar. Conversely, if human vision evolved in an environment in which shortcut recognition cues were available via indicator pixels, perhaps it would not have a shape bias.

This paper appears fundamentally flawed in its approach. The results are not informative about differences between human vision and CNNs, nor are they surprising to machine learning practitioners.